# Holmium-Containing Bioactive Glasses Dispersed in Poloxamer 407 Hydrogel as a Theragenerative Composite for Bone Cancer Treatment

**DOI:** 10.3390/ma14061459

**Published:** 2021-03-17

**Authors:** Telma Zambanini, Roger Borges, Ana C. S. de Souza, Giselle Z. Justo, Joel Machado, Daniele R. de Araujo, Juliana Marchi

**Affiliations:** 1Centro de Ciências Naturais e Humanas, Universidade Federal do ABC, Santo André 09210-580, SP, Brazil; telma.zambanini@ufabc.edu.br (T.Z.); roger.borges@aluno.ufabc.edu.br (R.B.); ana.galvao@ufabc.edu.br (A.C.S.d.S.); daniele.araujo@ufabc.edu.br (D.R.d.A.); 2Departamento de Bioquímica, Universidade Federal de São Paulo, São Paulo 04044-020, SP, Brazil; giselle.zenker@unifesp.br; 3Departamento de Ciências Biológicas, Universidade Federal de São Paulo, Diadema 04039-032, SP, Brazil; joel.machadojr@gmail.com

**Keywords:** bone cancer, brachytherapy, bioactive glass, holmium, hydrogel, theragenerative biomaterial

## Abstract

Holmium-containing bioactive glasses can be applied in bone cancer treatment because the holmium content can be neutron activated, having suitable properties for brachytherapy applications, while the bioactive glass matrix can regenerate the bone alterations induced by the tumor. To facilitate the application of these glasses in clinical practice, we proposed a composite based on Poloxamer 407 thermoresponsive hydrogel, with suitable properties for applications as injectable systems. Therefore, in this work, we evaluated the influence of holmium-containing glass particles on the properties of Poloxamer 407 hydrogel (20 *w*/*w*.%), including self-assembly ability and biological properties. 58S bioactive glasses (58SiO_2_-33CaO-9P_2_O_5_) containing different Ho_2_O_3_ amounts (1.25, 2.5, 3.75, and 5 wt.%) were incorporated into the hydrogel. The formulations were characterized by scanning electron microscopy, differential scanning calorimetry, rheological tests, and [3-(4,5-dimethylthiazol-2-yl)-2,5-diphenyltetrazolium bromide] MTT cell viability against pre-osteoblastic and osteosarcoma cells. The results evidenced that neither the glass particles dispersed in the hydrogel nor the holmium content in the glasses significantly influenced the hydrogel self-assembly ability (T_mic_ ~13.8 °C and T_gel_ ~20 °C). Although, the glass particles considerably diminished the hydrogel viscosity in one order of magnitude at body temperature (37 °C). The cytotoxicity results evidenced that the formulations selectively favored pre-osteoblastic cell proliferation and osteosarcoma cell death. In conclusion, the formulation containing glass with the highest fraction of holmium content (5 wt.%) had the best biological results outcomes aiming its application as theragenerative materials for bone cancer treatment.

## 1. Introduction

Bioactive glasses have been used as biomaterials for bone regeneration since the 70s when Larry L. Hench developed the first glass based on the 45SiO_2_-24.5CaO-24.5Na_2_O-6P_2_O_5_ system and able to chemically bond to the bone [1]. These bioactive glasses are applied in bone tissue regeneration because they promote the nucleation and growth of a hydroxyapatite-like layer on their surface after interacting with the body fluid, and these superficial chemical reactions are known as bioactivity, that is, the ability of a biomaterial to grow an apatite layer on their surface after interacting with the body fluid [2]. Besides their bioactivity, when these glasses interact with the body fluid, their surface is dissolved, and the dissolution products can promote specific biological responses [3]. For example, their dissolution products also act as cell signaling modulators in osteoblast-like cells, favoring the expression of growth factors and the induction of osteoblastic differentiation and proliferation, resulting in increased bone regeneration [4]. These biological properties make these glasses biocompatible; that is, they do not trigger a foreign-body response that could yield acute inflammation response [4,5,6,7]. Furthermore, although the first bioactive glasses were based on the aforementioned quaternary diagram, many other glass compositions are allowed, like borate [8] and phosphate glasses [9,10], and they all can be used in biological applications, as long as bioactivity and biocompatibility issues are met [11]. Although the initial application of bioactive glasses were intended for bone regeneration, over the years, other applications were suggested and reached in clinical studies, such as dental restoration [12], dentine hypersensitivity [13,14], drug delivery [5,15,16], tissue engineering [17,18,19], among others, including cancer treatment [20,21,22], which is the focus of this work.

Glasses lack long-range periodic ordering, which is a consequence of their amorphous nature, enabling almost all the elements from the periodic table can be incorporated in the glass structure, including rare-earth [23]. The incorporation of rare-earth ions in bioactive glasses confers them new properties like optical, nuclear, and magnetic, enabling their applications in contrast agents in magnetic resonance [24,25], imaging diagnostics [26,27], brachytherapy (internal radiotherapy) seeds [28,29], among others [30]. Particular attention should be given to brachytherapy applications of bioactive glasses. Brachytherapy still stands as a low-cost, effective therapy [31,32], and its clinical application is not based on bioactive glasses but instead on the bioinert 90-yttrium aluminosilicate glasses (^90^YAS, TheraSphere^®^, MDS Nordion, Ottawa, ON, Canada), which is used in the treatment of liver cancer by radioembolization [33,34].

However, since 2003, researchers have highlighted the possible applications of rare earth-containing bioactive glasses in bone cancer treatment, once these glasses could treat cancer and regenerate the bone lesion caused by the tumor. The first radioactive bioactive glass proposed for cancer treatment by brachytherapy was proposed by a Brazilian research group [35,36,37], who showed that bioactive glasses obtained by the sol-gel method, and based on the SiO_2_-CaO-Sm_2_O_3_ system, displayed nuclear properties similar to that of ^125^I seeds, which are the most common and standard materials used in the treatment of prostate cancer by brachytherapy. In summary, after producing the glass powder, it is submitted to neutron activation, yielding the production of ^153^Sm, which decays to ^153^Eu, and emits *β*-particles. Ionizing radiation comes from the decay of radioisotopes and can react with cells by direct or indirect mechanisms. Through the direct mechanism, they interact with the atoms of biomolecules, such as DNA (deoxyribonucleic acid), breaking down its structure. On the other hand, they can break down water molecules by the indirect mechanism, resulting in highly reactive oxidizing agents, which affect other vital cells, leading to cell death [38].

Since then, other researchers have studied bioactive glass compositions, aiming to incorporate other rare-earth and obtaining improved *β*-particle emission properties, allied with improved bioactivity [28,39,40,41,42,43]. The incorporation of holmium oxide in the structure of bioactive glasses has been highlighted by Nogueira and Campos [44], who showed that glasses based on the system SiO_2_-CaO-Ho_2_O_3_ have some advantages like shorter half-life (1,11 days) and lower quantities of holmium in the glass structure are need to obtain radiation properties similar to the ^125^I seeds used in prostate cancer. Also, Diniz et al. [45] showed that bioresorbable glasses based on the system SiO2-CaO-Ho_2_O_3_ do not yield cytotoxic response in in vivo applications in a mouse brain, even after 30 days implanted. All these finds suggest that holmium oxide may be prominent rare-earth for brachytherapy applications.

Recently, we showed that holmium ions in bioactive glasses based on the system SiO_2_-CaO-P_2_O_5_-Ho_2_O_3_ are responsible for stronger Si-O-Ho chemical bonds but lower glass connectivity, enabling proper glass dissolution kinetics to favor bioactivity [46]. However, given that the strong covalent bonds between holmium and adjacent non-bridging oxygens, holmium is leached into the body over slower dissolution kinetics, which guarantees the safety of using them as radioactive elements in the glass, once ^166^Ho will not be leached into the body at concentrations enough to damage healthy cells [46]. Because these glasses can; (i) regenerate the bone loss caused by cancer through their osteoinduction and bioactivity, (ii) besides treating the bone cancer by brachytherapy, these glasses can be defined as theragenerative biomaterials [47]. The word theragenerative comes from therapy and regenerative and is a new classification of biomaterials that can combine both properties in the same material and follow the same logic of theranostic materials [48].

Although studies of holmium-containing glasses have limited results concerning in vivo studies of brachytherapy, poly(l-lactic acid) microspheres containing holmium acetylacetonate have shown the prominent usage of ^166^Ho in cancer treatment [49]. For example, preclinical studies of these microspheres showed a 55% of local response rate, including complete and partial response, in the treatment of oral squamous cell carcinoma in cats [50]. Another study evaluated the efficacy of these microspheres in treating renal cancer in mice and showed that treated mice showed size tumor-control, while control mice showed expressive tumor growth [51]. Interestingly, in both approaches, the microspheres were administered intratumorally by suspending the microsphere in a Poloxamer 188 (2% *w*/*v*) soft-gel-like solution, creating a carrier system.

Similarly, the management of bioactive glasses for brachytherapy in the clinical practice can be facilitated by creating injectable, non-invasive systems able to deliver the glass particles into the therapeutic site. This strategy can be accomplished by producing composites materials based on bioactive glass particles dispersed in a hydrogel matrix, such as the Poloxamer above mentioned [52].

Poloxamers (PL) are non-ionic triblock copolymers, composed of two hydrophilic blocks of poly(ethylene oxide) (PEO) bonded through an intermediate hydrophobic block of poly(propylene oxide) (PPO), forming a PEOa-PPOb-PEOa structure, where a and b represent the chain size of each block and are responsible for modulating the properties of poloxamer [53]. Also, PL is a thermo-responsive polymer with different viscosity behavior according to the solution temperature and final polymer concentration: bellow micellization, the PL solution is liquid; between micellization temperature and sol-gel transition, the PL is a soft-gel; and above sol-gel transition, its behavior is like a hard-gel, conferring hydrogel properties [54,55,56]. Over the last years, poloxamers have been used in different applications like carriers of anti-cancer drugs, cancer diagnostics, gene transfection, an inhibitor of multidrug resistance effect, intranuclear-target delivery, and bioadhesives [57]. In relation to the application of PL as an inhibitor of multidrug resistance tumors, hydrogel-based drug delivery systems containing doxorubicin have even been stepped towards phase I and II clinical trials, showing effective tumor shrinkage in humans after treatment [58]. Therefore, PL has shown promising properties for cancer treatment applications.

In this context, this work aims to study the influence of different holmium content in bioactive glasses on the self-assembly ability and biological properties of PL 407-based hydrogel composites. The primary purpose of these composites is their application in bone cancer treatment, once they can act as theragenerative materials, treating bone cancer and regenerating the bone tissue. Furthermore, by using these composites as injectable systems, they become less-invasive and less traumatic to the patients, enabling their applications in multiple bone cancer sites.

## 2. Materials and Methods

### 2.1. Glass Synthesis by Sol-Gel Method

Five glass compositions based on 58SiO_2_-33CaO-9P_2_O_5_ (wt.%) system were studied: a parent glass and four compositions with the incorporation of holmium oxide (1.25 wt.%; 2.5 wt.%; 3.75 wt.% and 5 wt.%). The final compositions are shown in Table 1. These compositions have been studied by our research group, derived from the 58S bioactive glass [59]. They show high bioactivity and biocompatibility towards pre-osteoblastic cell (MC3T3) [46]. For the synthesis, tetraethylorthosilicate (TEOS—Sigma-Aldrich, St. Louis, MO, USA 99.99%, CAS # 78-10-8) and triethyl phosphate (TEP—Sigma-Aldrich, St. Louis, MO, USA, >99.8%, CAS # 78-40-0) were used as precursors of SiO_2_ and P_2_O_5_, respectively, and calcium nitrate tetrahydrate (Ca(NO_3_)_3_.4H_2_O—Sigma-Aldrich, St. Louis, MO, USA, >99.0%, CAS # 13477-34-4) and holmium nitrate pentahydrate (Ho(NO_3_)_3_.5H_2_O—Sigma-Aldrich, St. Louis, MO, USA, 99.9%, CAS #14483-18-2) were used as precursors of CaO, and Ho_2_O_3_, respectively. Glasses were obtained by modifying the sol-gel quick alkali route described in [46,60], which allows the obtainment of glass nanoparticles. Briefly, TEOS and TEP were hydrolyzed in a solution containing deionized water, ethanol, and 2 M HNO_3_ (Sigma-Aldrich, St. Louis, MO, USA, 70%, CAS #7697-37-2) in a 13.9:50:2 ratio. After 20 min, the nitrates were dissolved in this acid solution. Then, 10 mL of 2 M ammonia solution (Sigma-Aldrich, St. Louis, MO, USA, 20–30%, CAS # 1336-21-6) was quickly dropped (less than 2 s) into the acidic solution, causing a sol-gel transition. The collected gel was dried upon freezing drying and calcined at 550 °C.

### 2.2. Hydrogel Formulations and Morphological Characterization

PL 407 (Sigma-Aldrich, St. Louis, MO, USA, CAS # 9003-11-6) hydrogel was prepared by solubilization in buffer solution (20 mM Hepes buffer with 154 mM NaCl, at pH 7.4) in a 20% (*w*/*w*) polymer final concentration. Polymer dispersion in an ice bath under magnetic stirring (750 RPM) for twenty-four hours. For formulations preparation, bioactive glasses were dispersed in poloxamer solutions at 0.05 g/mL concentration and maintained in an ice bath under magnetic stirring for twelve hours. All formulations prepared are shown in Table 2. Both the glass particles and the hydrogel formulations had their morphology analyzed by scanning electron microscopy (SEM, FEI QUANTA 250, Hillsboro, OR, USA). The analyses were performed by applying an acceleration voltage of 0.8 kV and using a 10 mm working distance.

### 2.3. Differential Scanning Calorimetry (DSC)

Initially, samples were placed in hermetic aluminum pans and underwent three heating-cooling-heating cycles from 0 to 50 °C at a 5 °C/min rate, using an empty pan as a reference under nitrogen atmosphere. DSC experiments were performed in a DSC 214 Polyma equipment model (Netzsch, Wittelsbacherstraße, Selb, Germany). Thermodynamic parameters related to micellization, such as enthalpy (ΔH°, determined by the area under the endothermic peak on the heating cycle), Gibbs free energy (ΔG°), and entropy (ΔS°) were calculated, using Equations (1) and (2):ΔG° = R.Tmic · ln(x)(1)
ΔG° = ΔH° − Tmic · ΔS° (2)

R is the gas law constant (8.31 J × mol^−1^ × K^−1^), Tmic is the temperature for micellization in K, and x is the polymer concentration in mole fraction units.

### 2.4. Rheological Characterization

Two different experimental approaches analyzed the rheological behavior of samples: (1) At a constant frequency of 1 Hz with a temperature range from 5 to 60 °C; (2) frequency sweep analysis from 0.1 to 10 Hz at a constant temperature of 37 °C. All analyses were performed under a shear stress of 2 Pa. The rheological analyses were performed in triplicate using a Malvern Paranalytical^®^ (Malvern, Worcestershire, UK) KINEXUS rheometer, with a plate-plate geometry (40 mm) and a sample volume of 1 mL. Measurements as a function of temperature allowed to determine elastic modulus (G’), viscous modulus (G″), and viscosity (η) behavior, allowing the detection of the sol-gel transition temperature (Tsol-gel). Measurements as a function of frequency allowed to obtain values of elastic modulus (G′) and viscous modulus (G″) at 37 °C.

### 2.5. Cell Culture

MC3T3-E1 subclone 14 (American Type Culture Collection, ATCC, CRL-2594), an osteoblast precursor cell line derived from Mus musculus mouse calvaria, were grown in α-MEM medium (Alpha Modified Eagle’s Medium). MG63 (American Type Culture Collection, ATCC, CRL-1427), an osteosarcoma cell line derived from Homo sapiens, were grown in DMEM medium (Dulbecco’s Modified Eagle’s Medium). The medium was supplemented with 1% penicillin/streptomycin and 10% fetal bovine serum. The cells were grown in 25 cm^2^ flasks at an initial density of 5 × 10^4^ cells/mL and kept in a humidified atmosphere, 5% CO_2_, at 37 °C, and replaced every 72/96 h (MC3T3-E1) and 48/72 h (MG63). For the experiments, the cells were detached from flasks with trypsin, counted in a Neubauer chamber, and diluted in a supplemented medium to the concentration determined for the analysis. The cells were then plated in 96-well plates and maintained in a humidified atmosphere, 5% CO_2_, at 37 °C for 24 h before exposition to formulations’ conditioned mediums.

### 2.6. Conditioned Medium Preparation

Conditioned mediums were prepared by diluting 0.2 g/mL of each formulation in culture medium with 1% penicillin/streptomycin (α-MEM for MC3T3 cells and DMEM for MG63 cells) and incubated in a humidified atmosphere, 5% CO_2_, at 37 °C for 48 h. After this period, the medium was sterilized in a membrane filter (0.22 µm pore size) and supplemented with 10% fetal bovine serum. Each prepared extract was diluted in different concentrations in the respective medium (serial dilution) to obtained concentrations of 100%; 50%; 25%; 12.5% and 6.25%.

### 2.7. Cytotoxicity Assay

The effects of the studied systems on cell survival and proliferation were evaluated using 3-(4.5-dimethyl-2-thiazolyl)-2.5-diphenyl-2H-tetrazolium bromide (MTT) reduction test. MC3T3-E1 ATCC CRL 2594 (1 × 10^4^ cells/mL) and MG63 ATCC CRL 1427 (0.3 × 10^4^ cells/mL) cells were seed in 96 well plates and incubated in a humidified atmosphere, 5% CO2, at 37 °C. After 24 h, the cells were washed with phosphate-buffered saline (pH 7.4), and 100 μL of extract (in a diluted and non-diluted form) added to each well. After 72 h incubation, the cells were washed with phosphate-buffered saline (pH 7.4) and incubated for 2 h with MTT solution (0.5 mg/mL), prepared in medium without fetal bovine serum. Later, the absorbance was measured at λ = 570 nm with a plate reader’s aid, corresponding to each well. The values were expressed as percentages of MTT reduction concerning the negative control (cells incubated in the absence of extract) considered 100% [61].

## 3. Results

### 3.1. Physico-Chemical Characterization of Injectable Systems

The macroscopic photography of PL and PL-BG systems at room temperature is depicted in Figure 1A. PL is characterized as a transparent gel phase. Following the incorporation of glasses, the systems were white and opaque due to glass dispersion into the hydrogel. The white color was independent of the holmium content in the glasses. SEM images of glass particles are found in Figure 1B,C related to the BG, and BG5Ho glasses, respectively. SEM analysis of hydrogel formulations is presented in two different magnifications. PL sample (Figure 1D,G) containing a slightly irregular surface, structured as layers in a microscopic characterization. Similar morphology was observed for PL-BG (Figure 1E,H) and PL-BG5Ho systems (Figure 1F,I). The presence of glass did not alter the layer features of the hydrogel, although it increases the layer roughness. The absence of glass agglomerates suggests an adequate dispersion of the glasses into the hydrogel matrices. Moreover, regardless of the holmium content, all the glasses displayed the same influence on the hydrogel morphology.

Figure 2 shows the DSC thermograms for PL-BG5Ho-system, which is also representative once other systems had similar behavior. The thermogram is characterized by three cyclic curves: heating, cooling, and heating. Both heating cycles showed a broad endothermic peak related to Poloxamer micellization, while the cooling cycle showed a broad exothermic peak associated with a gel to sol transition. Besides thermoreversibility, other parameters are taken from DSC analysis, such as micellization temperature (T_mic_), micellization temperature onset (T_onset_), and micellization temperature endset (T_endset_) which are obtained from the aforementioned endothermic peak (Table 2). All formulations showed similar patterns: micellization onset was observed around ~10 °C and endset around 20.5 °C, with a micellization peak at 13.8 °C.

Regarding DSC analysis, thermodynamic parameters of micelle formation, such as enthalpy, entropy, and Gibbs free energy, calculated from the endothermic peak area, are also presented in Table 3. All formulations showed negative free Gibbs energy values, corresponding to an energetically favorable micellization process. The values of Gibbs free energy were similar for all formulations (around −9.35 kJ·mol^−1^), which agrees with the T_mic_ results reported (13.8 °C), once the Gibbs free energy depends only on the polymer fraction and T_mic_. Considering that the polymer concentration remains the same in all the systems, ∆G relies on T_mic_. In relation to ∆H and ∆S, the addition of glasses resulted in a small decrease in energy and work needed for micelle formation. For example, the ∆H of PL was 68.59 kJ·mol^−1^, while the formulations containing glasses showed values around 64 and 65 kJ·mol^−1^; the same pattern was noted regarding ∆S, that is, the value of 0.27 kJ·mol^−1^ for PL formulation, and values around 26 kJ·mol^−1^ for formulations containing glasses. However, these changes were not significant to suggest any influence of the glass particles on the thermodynamical properties of PL.

Figure 3 presents the two different rheological characterizations as a function of temperature for all formulations: (a) Variation of elastic G′ and viscous G″ moduli (Figure 3A); (b) the viscosity behavior as a function of the temperature (Figure 3B). These analyses contribute to understanding the influence of glasses on the sol-gel transition, which is the next step after micellization due to the micelles’ further self-assembly into crystalline supramolecular arrangements in response to temperature rising. Concerning the influence of the temperature on G′ and G″, at low temperatures, the values of G′ and G″ are shallow, with G″ greater than G′. At about 20 °C, the value of both moduli increases sharply, with G′ exceeding G″ values. The interception between G′ and G″ corresponds to the sol-gel transition (Table 3). The sol-gel temperatures were similar for all formulations, without significant difference after analysis by Tukey test (*p* < 0.05). The G″ values are higher for compositions containing glasses, independent of holmium content.

In relation to the influence of temperature on viscosity profiles (Figure 3B), all formulations showed similar behavior of G’ modulus, that is, values in a 10^−1^ mPa·s of magnitude before the sol-gel transition, followed by a sharp increase during the sol-gel transition (around 20 °C), reaching viscosity values in the magnitude of 10^6^ mPa·s. The viscosity values obtained at different temperatures (10, 25, and 37 °C) are shown in Table 4. Overall, before the sol-gel transition (10 °C), the presence of glasses in the formulations led to increased viscosity. However, the holmium content in the glasses did not influence their viscosity, which means no significant difference among all the formulations containing glasses. Besides, above the sol-gel transition, 37 °C, those formulations containing glasses displayed lower viscosity than the PL one.

For PL-based formulations, viscoelastic behavior is usually associated with G′ values 15 or 20-fold higher than G″, as Oshiro et al. [30] discussed. When the formulation is in the hard-gel state, above 37 °C, a rheological characterization about gel stability is desired, which is performed through the analysis of G′ and G′ values at 37 °C as a function of the frequency, as shown in Figure 4A. All formulations displayed a hard-gel behavior with G′ values higher than G″ at different frequencies. The addition of glasses in the systems resulted in higher G″ values, which can be attributed to the disruption on micelles self-aggregation following the glasses addition. Moreover, the glasses incorporation decreased the G′/G″ ratio compared to PL formulation, even though the viscoelastic behavior of all formulations are maintained (Figure 4B). Viscoelastic behavior is usually associated with G′ values 15 or 20-fold higher than G″, as Oshiro et al. [30] discussed.

### 3.2. Biological Characterization of Injectable Systems on the Viability of MC3T3-E1 Osteoblastic and MG-63 Bone Cancer Cells

First, we checked the effect of a conditioned medium containing different formulation components on osteoblastic MC3T3-E1 cell viability. Treatment with formulations 100% concentrated increased the viability of MC3T3-E1 cells compared to control cells treated only with conditioned medium free of formulation components, suggesting that all tested formulations were not cytotoxic but rather increased cell proliferation (Figure 5A). In contrast, upon treatment with formulations <100% concentrated, the percentage of viable cells decreased in a concentration-dependent manner, returning to nearby control levels (Figure 5A). To gain insight into the influence of distinct components present in the formulations that could account for the increased cell viability effect seen upon treatment with formulations 100% concentrated, a statistical analysis was performed for this specific condition. As shown in Figure 5B, while formulation containing exclusively PL increased MC3T3-E1 cells viability, the addition of un-doped glass (BG) in PL (PL-BG formulation) counteracted this effect. On the other hand, when holmium was added to the glass structure, cells treated with formulations containing holmium neutralized the effect of BG, showing cell viability levels similar to cells treated with formulations containing PL only (Figure 5B).

In contrast to osteoblastic MC3T3-E1 cells, treatment of MG-63 cancer cells with formulations 100% concentrated showed an expressive decrease in cell viability, mainly when BG was added in the PL system (PL-BG formulation) (Figure 5C,D). Although a slight decrease in the viability of MG-63 cells was observed in formulations containing PL only, the addition of BG to the system significantly enhanced this effect, suggesting that MG-63 but no MC3T3-E1 cells are cytotoxically sensitive when exposed to formulations containing BG (Figure 5C,D). Interestingly, the addition of holmium in the glass structure counteracted the cytotoxic effect of BG-containing formulations (Figure 5D). Furthermore, when MG-63 cells were treated with formulations <100% concentrated, cell viability was gradually rescued to nearby control levels compared to treatment with formulations 100% concentrated.

## 4. Discussion

The poloxamer 407-based hydrogel is a thermoreversible and biocompatible system, making it a suitable material for minimally-invasive injectable systems aiming at tissue regeneration [62]. This study intended to incorporate holmium-doped bioactive glasses in the hydrogel structure of poloxamer 407, targeting bone cancer treatment applications. Because the holmium content in the glasses is responsible for modulating the nuclear properties for brachytherapy applications, we developed a series of glasses with different holmium content, which could be used for bone cancer treatment following the dose rate designed for each patient. Our research group has already characterized these glasses regarding their dissolution, bioactivity, and cytotoxicity to MC3T3 cell lineage [46]. In this previous work, we also showed that holmium was incorporated in the glass network without any devitrification. Therefore, this work focused on understanding the influence of the glasses on the Poloxamer 407 properties, and it was desired to study the role of holmium content on those properties.

Poloxamer 407 forms hydrogel due to two phenomena: (i) micelle formation ability; (ii) the micelles’ capability to self-assemble into crystalline supramolecular arrangements related to the sol-gel transition [63]. PL chains can self-assemble into micelles when the polymer is found in an aqueous solution at a critical micellar concentration (CMC) and critical micellar temperature (CMT). After self-assembling in micelles, if the temperature of the colloidal solution keeps rising, the micelles suffer another self-assembly into supramolecular crystalline arrangements, increasing the solution viscosity, yielding a sol-gel transition. In this sense, while the DSC characterization brings new insights about the influence of glass nanoparticles on the micelle formation process, the rheological characterization helps to relate the influence of glass nanoparticles on sol-gel transition, hydrogel viscosity, and systems stability. Therefore, taken together, the thermal and rheological characterization gives a broader overview of the influence of glass particles on the whole process of hydrogel formation. Before discussing the thermal and rheological properties of the studied formulations, a morphological analysis of glass particle distribution in the hydrogel is needed, considering that heterogeneities may affect these properties, mainly hydrogel structural organization [64].

The macroscopic (photography) and microscopic (SEM images) characterizations (Figure 1) showed that the glasses were well dispersed into the poloxamer 407 hydrogel matrix. In the SEM images, no glass aggregates were expressly noted, suggesting a good dispersion of the glass particles in the hydrogel. Then, the DSC and rheological characterization were interpreted as an analysis of a homogeneous system due to the inexistence of micrometric agglomeration of glass particles.

The DSC results (Figure 2) evidenced that poloxamer thermoreversibility was maintained after the glass addition, which was characterized by the presence of an endothermic peak in the heating curves, even after a heating-cooling-heating cycle. Also, the presence of holmium in the glasses neither influenced thermoreversibility nor T_mic_. In contrast, the addition of glasses in the poloxamer matrix caused changes in the energy needed for micellization, despite not changing the T_mic_. Formulations containing glasses showed decreased ∆H and ∆S values, implying less energy needed for micellization self-assembly. In other words, the glasses do not change the CMT (critical micelle temperature) but instead lowered the energy required for micellization. This may be related to ions like calcium, holmium, and mainly phosphates leached from the glass during the homogenization of formulations. These ions are likely to act as water structurer, favoring the hydrogen bonds between water molecules rather than between water and poloxamer, which, in turn, favors micellization [65,66]. Considering that these ions come from glass dissolution, the different content of holmium in the glass structure was supposed to influence the release of ionic dissolution products. Previous work carried out by our research group [30,46] showed that bioactive glasses containing rare earth display slower dissolution kinetics, caused by stronger chemical bonds between rare earth and non-bridging oxygens [42]. However, considering poloxamer-based systems, when Ho-glasses are compared with the undoped-glass, it seems that the influence of holmium in glass dissolution was not enough to lead to significant changes in the enthalpy and entropy micellization of poloxamer.

Once the influence of glasses on the micellization behavior of our proposed injectable systems was clarified, the next step was to understand the influence of glasses on the sol-gel transition: the self-assembly of micelles into crystalline supramolecular structures. The rheology characterization of G′ and G″ as a function of the temperature (Figure 3A) showed that the addition of glasses does not play a significant role on T_gel_, but displayed a significant influence on viscosity (Figure 3B). Bellow the sol-gel transition (~20 °C), the formulations behave like a Newtonian fluid. In this case, glasses increased the viscosity because the friction between the glass particle surface and adjacent fluid layers increases the shear resistance [67]. However, above the sol-gel transition, all formulations behave like non-Newtonian shear-thinning fluid because the supramolecular structure of hydrogel acts as a colloidal solution [68,69]. Furthermore, those formulations containing glasses showed lower viscosity than the PL one above the sol-gel transition. This find may be caused by a maintained dispersion of glass particles in the hydrogel even under shear stress, favoring shear-thinning flow. Also, given that this effect is somewhat related to glass particle size than its composition, all the formulations containing glasses behave similarly regardless of the holmium content in the glass composition. Furthermore, considering that the sol-gel transition was observed near 20 °C, in clinical practice, the formulations would be manipulated at low temperature, using an ice bath, for example, before injecting the formulations in the cancer site. This procedure aims to guarantee proper low viscosity for intratumor administration of the formulations using a syringe for a minimally invasive procedure.

Moreover, although the addition of glasses in the formulations increased the shear-thinning behavior of the hydrogel under shear stress, all the formulations remained stable at 37 °C, as evidenced by the rheological characterization (Figure 4). Even though the PL formulation shows G′/G″ ratio 8-fold higher than the glass-containing formulations, all of them are found in the same unit of magnitude (10^6^ mPa·s^−1^). Also, G′ is always between 15 and 20-fold higher than G″, which is typical of stable viscoelastic colloidal solution [55,70]. Therefore, the shear-thinning behavior of the formulations does not influence their stability over different frequencies, making these formulations promising materials for applications as injectable systems.

The MTT results showed that our formulations 100% concentrated were selectively cytotoxic to bone cancer cells (MG63 cells) while stimulating osteoblast-like cell proliferation (MC3T3-E1 cells). In previous work, we studied the MC3T3-E1 cell viability through indirect contact with holmium-doped glasses, using conditioned mediums at different concentrations, and observed that the glasses used in that study are biocompatible [46]. We also observed that glasses containing more holmium (BG5Ho composition) stimulated pre-osteoblastic cell proliferation. In this current work, we also observed that the formulations containing BG5Ho glass showed higher MC3T3-E1 cell proliferation levels while promoted cytotoxic effects on MG63 cells. All formulations displayed a cytotoxic effect on MG63 cells, but the PL-BG5Ho formulation showed the highest cytotoxic effect among those formulations containing holmium.

A note of caution is due here. It is very likely that the cytotoxic effect of these formulations is derived from dissolution products from the formulations after conditioning in the culture medium, and are not derived from any brachytherapy effect, because the glasses were not neutron activated. Also, we addressed glass dissolution issues to the lack of dose-dependent effect of holmium content in the glasses in the cytotoxicity results obtained in this work. In previous works [30,42,46], we showed that the addition of rare-earth on glass structure leads to decreased glass network connectivity at the same time that Si-O-RE (RE = rare-earth) bonds show a higher chemical bonding energy than Si-O-Si bonds. Therefore, there is a counterbalance between these two effects, yielding no significant changes in glass dissolution. However, when a significant amount of holmium is added to the glass structure, which is the case of the BG5Ho glass, the effect of the rare-earth on enhanced glass dissolution becomes more prominent. That is why the formulations containing BG5Ho glasses show different patterns than other glasses containing holmium. Moreover, the high selectivity cytotoxicity to MG63 cells observed in the composition PL-BG encouraged further studies to evaluate its application in cancer treatment, even though it does have exciting properties for brachytherapy applications. All the discussion concerning the cytotoxic effect displayed by the glasses will be conducted considering their dissolution products.

First, we will discuss the effect of rare earth on bone regeneration, and later we will continue to discuss the effect of calcium released from bioactive glasses on bone cancer cells. Rare-earth ions, including Ho^3+^, have ionic radii similar to Ca^2+^, acting either as agonist or antagonist of Ca^2+^ sites in biochemical pathways, and can stimulate pre-osteoblastic cell proliferation [71]. Recently, Zhu et al. [72] showed that Gd^3+^ ions leached from bioactive glasses could stimulate protein expression related to the Wnt signaling pathway, which is related to osteogenic differentiation of human bone marrow-derived mesenchymal stem cell. In this mechanism, Gd^3+^ ions would be related to increased expression of protein kinase B (Akt) and glycogen synthase kinase β (GSK3β), which together form the Akt/GSK3β pathway. In this sense, Akt inactivates GSK3β through phosphorylation on Ser9 of GSK3β; the GSK3β is part of the “destroy complex,” which is a complex responsible for degrading *β*-catenin, but its inactivation by phosphorylation also inactivates the destroy complex, yielding in increased *β*-catenin concentration in pre-osteoblastic cells. In turn, *β*-catenin is a co-factor involved in the gene expression of osteogenic factors; thereby, its increased concentration in the cytoplasm leads to enhanced osteogenic differentiation. In the same work, Zhu et al. also showed that Gd-bioactive glasses were able to improve in vivo bone regeneration, increased alkaline phosphatase (ALP) activity, and induce higher deposition of mineralized matrix. Altogether, these results highlights strong evidences of a possible biochemical pathway activated by rare earth, and related to promoted bone regeneration. A similar effect of Ho^3+^ on MC3T3-E1 cells might not be excluded, and can be related to the increased cell proliferation found in the cell viability test of the PL-BG5Ho formulation.

In relation to the effect of dissolution products from bioactive glasses on cancer cells, Sui et al. [73] showed that Ca^2+^ ions play a crucial role in the cytotoxic effect from bioactive glasses. The authors suggested dissolution products from bioactive glasses activate transient receptor potential channels and calcium-sensing receptors on tumor cells, favoring calcium influx. Inside the cancer cells, Ca^2+^ is involved in regulating the capain-1, a Ca^2+^-dependent cysteine protease involved in the apoptosis mediated by caspase-3, a key regulator of apoptosis. In this mechanism, calpain-1 cleaves the Bcl-2, which regulates the cellular homeostasis and represses apoptosis, finally yielding in the caspase-3-mediated irreversible apoptosis. The proposed mechanism was confirmed by transmission electron microscopy and Western blot. The same mechanism was not found in healthy cells, and the author speculated that healthy cells could shut down Ca^2+^ influx to protect themselves against damage, while cancer cells keep activating Ca^2+^ influx due to their vigorous metabolism. The proposed mechanism can explain why our PL-BG formulation showed higher cytotoxicity against MG63 cells than the other formulations since the BG glass has a higher calcium content than the other glasses.

However, not only do the glasses display a cytotoxic effect on MG63 cells, but also poloxamer. The copolymer is associated with an anti-cancer effect through a mechanism that involves higher fluidity of cell membrane and decreased ATP availability [66]. This find from the literature explains why our PL formulation showed a cytotoxic effect on MG63 cells, even though it does not contain bioactive glass. The selective cytotoxic effect of all formulations on MG63 cells while increasing the viability of MC3T3-E1 cells is an exciting and positive result. Based on our MTT results, it is possible to hypothesize that these formulations containing glasses may play a dual effect inducing cancer death while favoring osteoblastic cell differentiation and bone regeneration in vivo. In this regard, these formulations may present potential as theragenerative materials, a new class of biomaterials [47] that can perform therapy and regenerate tissues simultaneously. This new class of biomaterials allows simplification of medical procedures in the clinical practice due to its multifunctionality since the same material combines therapeutic and regeneration effect after implanting and may stand as the future of biomaterials together with theranostics materials [48]. Future in vitro studies will clarify the effectiveness of these formulations inducing osteogenic differentiation and bone regeneration.

## 5. Conclusions

Formulations made of holmium-containing glasses dispersed in a poloxamer 407 hydrogel were obtained, and the influence of glass particles on self-assembly ability and biological properties of the hydrogel was evaluated. Calorimetric results evidenced that glass ions leached from the glasses favored poloxamer micellization but did not influence sol-gel transition, as confirmed by rheological characterization. However, glass particles in the hydrogel yield a slight diminishment in viscosity, although all the studied formulations show suitable properties for applications as injectable systems. Furthermore, the biological characterization showed that the formulations could selectively favor pre-osteoblastic cell proliferation and cause the death of osteosarcoma cells. The formulation based on poloxamer incorporated with glass containing 5 wt.% of Ho_2_O_3_ showed the most prominent properties for applications as theragenerative material for bone cancer treatment.

Moreover, we did not observe significant influence of holmium content on the properties of the formulations, which is considered an advantage of such materials. The amount of holmium in the glasses determines the radiation dose of the brachytherapy in future studies. Then, the formulations can be developed with the same properties but with different radiation doses, which can be clinically used for different proposals. Further in vivo studies should be performed to evaluate neutron-activated ^166^Ho effect on tumor regression and bone regeneration.

## Figures and Tables

**Figure 1 materials-14-01459-f001:**
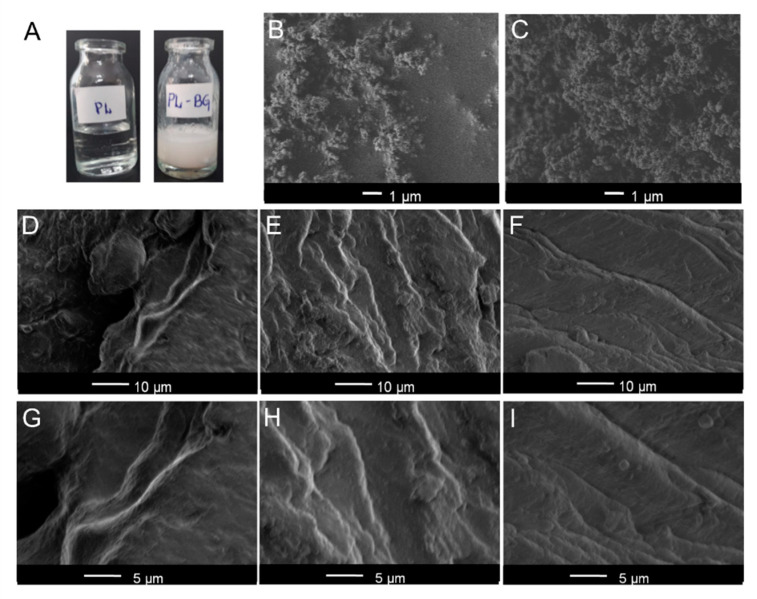
Morphological characterization (**A**) Photograph of Poloxamer (PL) and binary poloxamer-glasses (PL-BG) system; (**B**,**C**) morphology of glass particles: (**B**) BG; and (**C**) BG5Ho; (**D**–**F**) Typical scanning electron micrographs of PL (**D**); PL-BG (**E**); PL-BG5Ho (**F**); (**G**–**I**) SEM images at higher magnification of PL (**G**); PL-BG (**H**); and PL-BG5Ho (**I**).

**Figure 2 materials-14-01459-f002:**
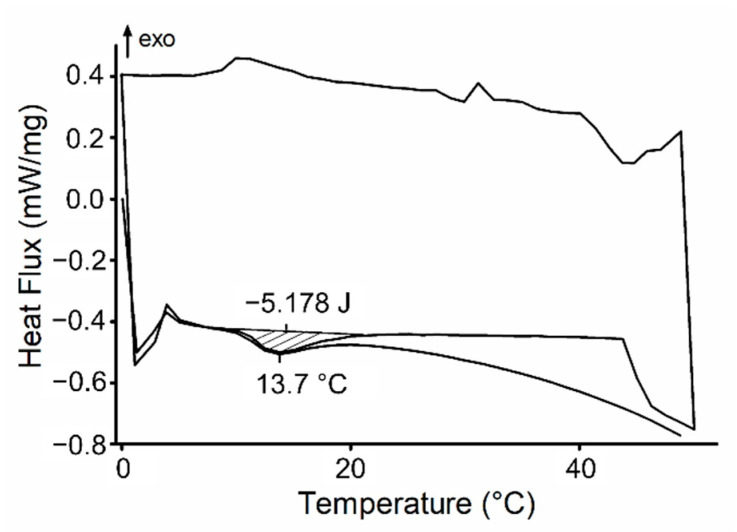
Behavior of differential scanning calorimetry of the PL-BG5Ho formulation. All the other formulations exhibited similar patterns.

**Figure 3 materials-14-01459-f003:**
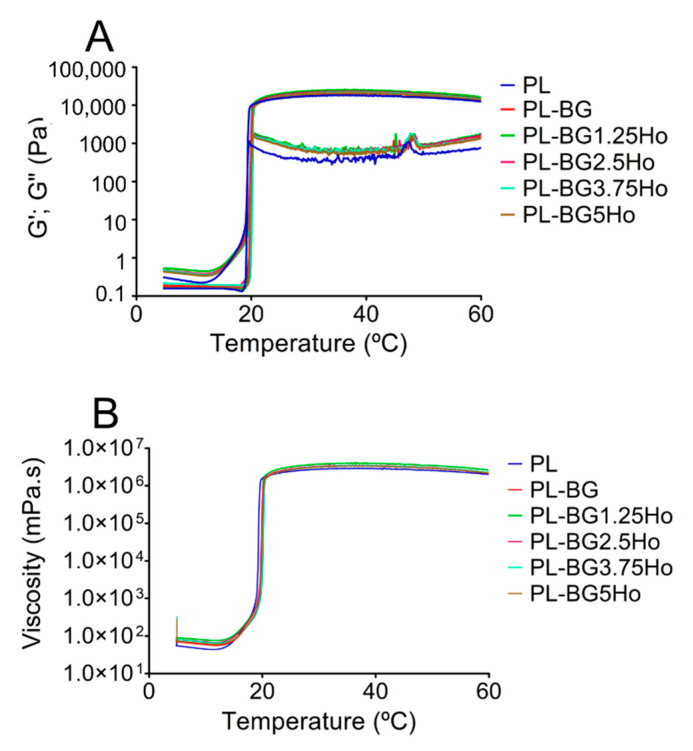
Rheological behavior as a function of the formulations’ temperature: (**A**) G′ and G″ values; (**B**) viscosity values.

**Figure 4 materials-14-01459-f004:**
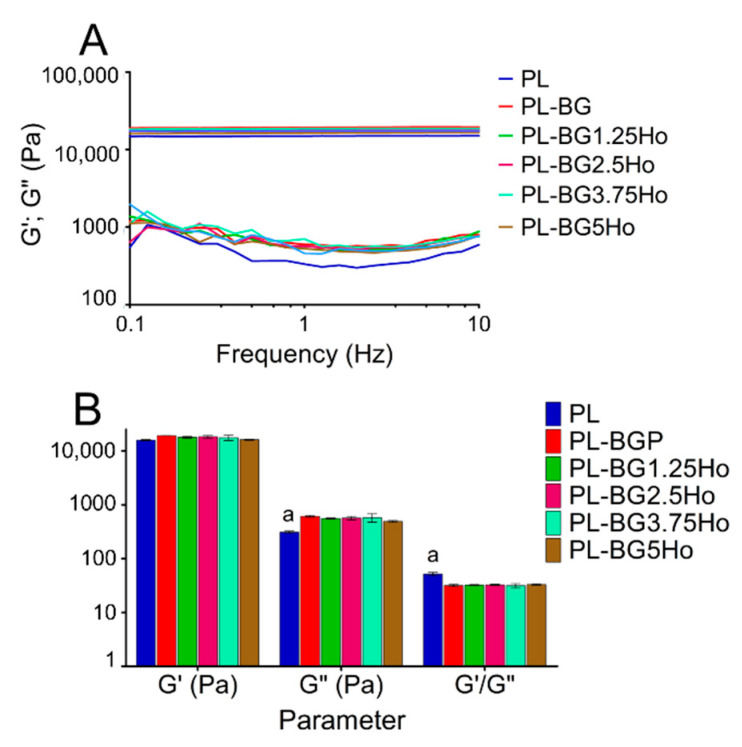
Rheological characterization as a function of frequency: (**A**) G′ and G″ values in function of frequency, at 37 °C, for the studied formulations. (**B**) G′; G″ and G′/G″ relationship values at 1 Hz frequency, and temperature of 37 °C. The columns highlighted with (a) differ statistically from the other groups, according to the Tukey test (*p* < 0.05),

**Figure 5 materials-14-01459-f005:**
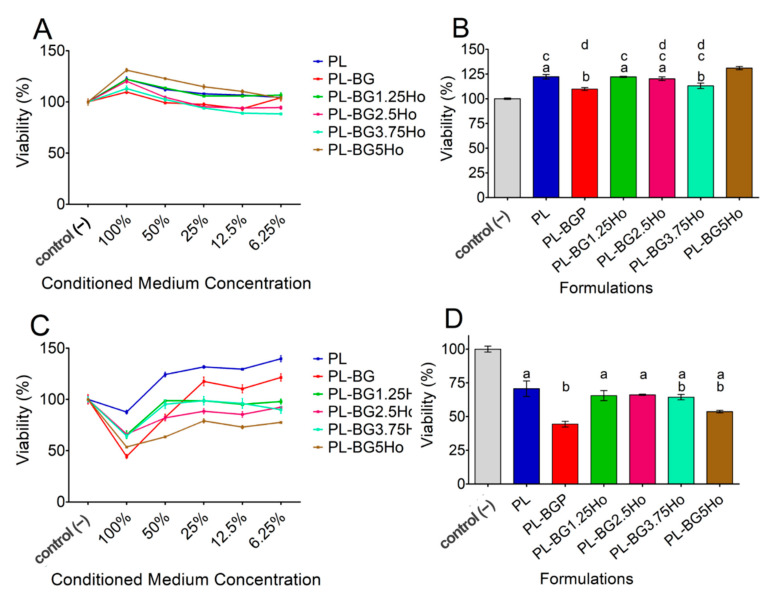
Cell viability behavior after 72 h using different cells exposed to conditioned medium of the studied systems in different concentrations of the extract by the MTT reduction analysis: (**A**) and (**B**) MC3T3-E1 cells; (**C**,**D**) MG63 cells; (**B**) and (**D**) comparison of cell viability response in a 100% extract concentration. According to the Tukey test, columns marked with the same symbol (a, b and/or c) do not differ (*p* < 0.05).

**Table 1 materials-14-01459-t001:** Glass compositions evaluated in this study (wt.%).

Nomenclature	SiO_2_	CaO	P_2_O_5_	Ho_2_O_3_
BG	58.00	33.00	9.00	-
BG1.25Ho	57.28	32.59	8.89	1.25
BG2.5Ho	56.55	32.18	8.78	2.50
BG3.75Ho	55.83	31.76	8.66	3.75
BG5Ho	55.10	31.35	8.55	5.00

**Table 2 materials-14-01459-t002:** Formulation compositions evaluated in this study.

Poloxamer	Bioactive Glass	Formulation
PL407	None	PL
	BG	PL-BG
	BG1.25Ho	PL-BG1.25Ho
	BG2.5Ho	PL-BG2.5Ho
	BG3.75Ho	PL-BG3.75Ho
	BG5Ho	PL-BG5Ho

**Table 3 materials-14-01459-t003:** Results from differential scanning calorimetric (DSC) analysis related to micellization process: Temperatures (Tonset, Tmic, Tendset), Enthalpies (ΔH°), Free Energies (ΔG°), and Entropies (ΔS°) of micelle formation of the studied injectable systems.

Formulation	T_onset_ (°C)	T_mic_ (°C)	T_endset_ (°C)	ΔH° (kJ·mol^−1^)	ΔG° (kJ·mol^−1^)	ΔS° (kJ·mol^−1^·K^−1^)
PL	9.5	13.8	20.0	68.59	−9.35	0.27
PL-BG	10.0	13.8	20.5	65.31	−9.35	0.26
PL-BG1.25Ho	10.5	13.8	20.5	64.84	−9.35	0.26
PL-BG2.5Ho	10.5	13.8	20.4	64.35	−9.35	0.26
PL-BG3.75Ho	10.7	13.9	20.5	59.43	−9.35	0.24
PL-BG5Ho	10.5	13.7	20.0	65.24	−9.34	0.26

**Table 4 materials-14-01459-t004:** Sol-gel transition temperature (T_gel_) and viscosity values at different temperatures after rheological analysis.

		Viscosity Values (mPa·s) at Different Temperatures		
Formulation	T_gel_	10 °C	25 °C	37 °C
PL	19.64 ± 0.72	43.68 ± 1.32	(23.25 ± 0.07) × 10^6^	(27.89 ± 0.10) × 10^6^
PL-BG	19.92 ± 0.30	67.04 ± 8.19	(2.70 ± 0.11) × 10^6^	(3.38 ± 0.09) × 10^6^
PL-BG1.25Ho	19.87 ± 0.12	73.52 ± 13.60	(3.09 ± 0.20) × 10^6^	(3.87 ± 0.23) × 10^6^
PL-BG2.5Ho	20.07 ± 0.28	65.13 ± 7.94	(2.87 ± 0.05) × 10^6^	(3.55 ± 0.13) × 10^6^
PL-BG3.75Ho	20.08 ± 0.47	61.68 ± 11.14	(2.63 ± 0.21) × 10^6^	(3.30 ± 0.17) × 10^6^
PL-BG5Ho	20.05 ± 0.25	62.14 ± 4.83	(2.84 ± 0.15) × 10^6^	(3.50 ± 0.25) × 10^6^

## Data Availability

The data presented in this study are openly available Mendeley Data, V1, at DOI 10.17632/82tdjvsrwk.1.

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
