# Peer review of "Holmium-Containing Bioactive Glasses Dispersed in Poloxamer 407 Hydrogel as a Theragenerative Composite for Bone Cancer Treatment"

_materials, 2021, doi:10.3390/ma14061459_

Round 1

Reviewer 1 Report

This manuscript describes the development of hydrogel composites for cancer treatment using poloxamers incorporating holmium-doped bioactive glasses. 

The authors have designed a good set of experiments and their conclusions are well supported by their results. Furthermore, the authors present how their work may lead to in vivo applications. 

My only comment is related to cell culture experiments. Have the authors considered running cell viability experiments in the presence of the hydrogel (as opposed to using the conditioned media) so that the results are more relevant to the in vivo situations? 

Reviewer 2 Report

Comments to Authors on the Manuscript Number: 1112317

The paper “Holmium-Doped Bioactive Glasses Dispersed in Poloxamer 407 Hydrogel as a Theragenerative Composite for Bone Cancer Treatment”, by T. Zambanini, R. Borges, A.C.S. Souza, G.Z. Justo, J. Machado Jr, D.R. de Araujo and J. Marchi, approaches a composite system based of holmium-containing glass particles and Poloxamer 407 hydrogel with the aim of proposing a new and performant solution that simultaneously responds to the requirements of cancer therapy and bone regeneration. The final samples were characterized from morphological, thermal, rheological and cellular point of view.

The overall idea is attractive, constituting a starting point for producing multifunctional materials that could improve the healing of bone cancer, but the results represent just a preliminary analysis of their clinical potential, with a number of gaps in terms of demonstration. The article presents several weaknesses, as follows:

  1. Title: the term “doped” is not appropriate in this case and I suggest the use of the word “containing” all over the manuscript.
  2. Abstract: - it is too long and needs to be revised in order to shorten it;

- the first phrases that create a theoretical background must be removed or transferred to the Introduction section;

- almost no numerical data are included in this section;

- please clarify the meaning of the last part from the phrase “The results evidenced that glass particles dispersed in the hydrogels favored the poloxamer’s micellization but did not influence sol-gel transition.” (since the sol-gel method was employed to produce glass particles, confusions can occur; this is applicable for the whole text).

  1. Keywords: too many and some of them too long.
  2. Introduction: - the information about bioactive glasses is limited and should be completed with other details, sustained by new references;

- more examples regarding holmium use are necessary, together with some explanations on the functioning mechanism in cancer treatment;

- citing several case studies regarding injectable systems and poloxamers is required;

- most of this section is pure theory, which is not recommended for a research article.

  1. Materials and methods: - please explain the selection of that particular oxide system and the reasons behind the ratios established between oxides;

- since you employ ammonia solution and change the general pH to a basic one, are you sure that the applied method is sol-gel? (it seems more like a precipitation route or a combined one);

- the term “hydrogels” appears several times in the manuscript, which is not correct since only one hydrogel was approached;

- no information is provided about the equipment used for the morphological investigation.

  1. Results: - the SEM images should be edited to a certain extent in order to eliminate the lower black band and keep only the scale bar;

- how come no image of the glass particles is included?

- the quality of Figure 2 is very poor and I suspect that the graph was taken directly from the equipment, without other efforts for plotting in a professional way;

- the statement “Regarding ΔH and ΔS, the addition of glasses resulted in a small decrease in energy and work needed for micelle formation, suggesting that the glasses favored the micellization.” is forced since the decrease is very small to validate such a conclusion;

- in Figures 3, 4 and 5 it is difficult to discern between different shades of blue and the resolution of the entire figures is not appropriate;

- the graphs from Figure 5 are repeated for three times.

  1. Discussion: - how to observe glass agglomerates in the hydrogel when the magnification of the SEM images is 2.000? (if they are nanoparticles, it is impossible to visualize even aggregates);

- there is no certainty regarding the glassy character of the particles (no investigation was performed in this regard);

- the incorporation of holmium is questionable in the absence of evidences to that effect;

- overall, the variations of the parameters are debatable since the increase of holmium content does not generate an expectable proportional influence all the time.

  1. Conclusions: - please add one or two supplementary phrases regarding the use of such materials, their limitations and the future improvements.
  2. References: - more recent references are required.

In conclusion, the paper “Holmium-Doped Bioactive Glasses Dispersed in Poloxamer 407 Hydrogel as a Theragenerative Composite for Bone Cancer Treatment”, by T. Zambanini, R. Borges, A.C.S. Souza, G.Z. Justo, J. Machado Jr, D.R. de Araujo and J. Marchi, can be published in Materials (MDPI) after a major revision.

Reviewer 3 Report

Dear Editor,

The article titled “Holmium-doped bioactive glasses dispersed in Poloxamer 407  hydrogel as a the regenerative composite for bone cancer treatment” was reviewed. This work aims to study the influence of different holmium content in bioactive glasses on the self-assembly ability and biological properties of PL 407-based hydrogel composites.

This study is well designed and conducted.  However, before publications authors have to consider the following comments.

  1. Title.

Instead of  “ragenerative”  write  “ regenerative”

  1. Abstract

Why do authors   mentioned about 166Ho in the Abstract although no information about  using 166Ho  appeared in  the text of the article.

  1. Results

-Fig.3.   Viscosity values should be presented in the following manner: 1x101 mPa· s,  1x102 mPa· s , etc.

- Table 3. Instead of spelling viscosity value as (23.25±0.07)*106 mPa· s   write as (23.25±0.07) ·106 mPa·s.

-  Fig.5. Pay attention to the fact that each figure underwent 3-times repetition.  

  1. Discussion

-It is written that “ All formulations displayed a cytotoxic effect on  MG63 cells, but the PL-BG5Ho formulation showed the highest cytotoxic effect among  those formulations containing holmium”.

However, from Fig.5 (c and d) highest cytotoxic effect on MG63 cells showed PL-BG composition and not PL-BG5Ho.  In this regard question is why not consider holmium – free glass dispersed in Poloxamer 407 hydrogel as potential regenerative composite for bone cancer treatment?

-Authors should discuss absence of   a concentration-dependent effect of holmium – doped glasses  on  MC3T3-E1 and MG63 cells viability.

-Authors should explain how composites are supposed to be applied   in a medical practice considering that sol-gel transition occurred at about 20 °C followed by  a drastic viscosity rise. 

Yours sincerely,

Reviewer

Reviewer 4 Report

General comment

This paper seems to be well structured. The main hypothesis is clear and the experimental study well defined.

Line 18: What is meant by 166Ho

Line 120 and following: Please add the order numbers of the used chemicals, that exact the same chemicals can used by other groups which wants to redo the experiments (with other background)

Line 128: What means quickly? Please add a correct value (10 seconds, 1 minute …)

Line 129: Did you measure the size of the resulting nano particles?

Line 136 + 138: Which RPM was used for the magantic stirring? It makes a difference if you used 100  or 4000 RPM

Line 159: Which diameter had the used plate of the rheometer?

Figure 1: Which size has the scale bar - is not readable if you print the paper

Figure 1: Which magnification is shown in this Figure? The SEM parameters, acceleration voltage, horizontal field width are missing: are B, C, D in same magnification?

Line 167 + 168: ATCC numbers of the different cell lines are missing

Line 238: „…the values of gibbs free energy were similar for all formulations with values of …. which  agrees with …“ please add some values (the text should be understandable without watching into the table/figure and vice versa)

Figure 2 … Figure 5: Text in Figures  is too tiny – is not readable if you print the paper

Line 350: If the nano particles has an influence on the micelle formation – did you measure the „size effect“ of the nano particles? 

Round 2

Reviewer 2 Report

Even though I still have some reservations regrading the characterization of the glasses considered separately, as well as the value of some results, I agree with the publication.

Reviewer 3 Report

Dear Editor,

The authors responded to all my questions and made the necessary changes to the manuscript. The manuscript in its revised form can be accepted to publication.

The Reviewer